# A comparative case study of the accommodation of students with disabilities in online and in-person degree programs

Chris Mead[1]*, Chad Price[2], Logan E. Gin[3], Ariel D. Anbar[1,4], James P. Collins[5], Paul LePore[6], Sara E. Brownell[5,7]

1 Center for Education Through Exploration, School of Earth and Space Exploration, Arizona State University, Tempe, Arizona, United States of America, 2 Student Accessibility and Inclusive Learning Services, Educational Outreach and Student Services, Arizona State University, Tempe, Arizona, United States of America, 3 Sheridan Center for Teaching and Learning, Brown University, Providence, Rhode Island, United States of America, 4 School of Molecular Sciences, Arizona State University, Tempe, Arizona, United States of America, 5 School of Life Sciences, Arizona State University, Tempe, Arizona, United States of America, 6 School of Social and Family Dynamics, The College of Liberal Arts and Sciences, Arizona State University, Tempe, Arizona, United States of America, 7 Research for Inclusive STEM Education Center, Arizona State University, Tempe, Arizona, United States of America

* Chris.Mead@asu.edu

**Data Availability Statement:** The data analyzed in this study describe students' individual disability statuses. They also include enough demographic information to make re-identification of individuals

## Abstract

Fully online degree programs are an increasingly important part of the higher education ecosystem. Among the many challenges raised by the growth of fully online courses and degree programs is the question: Are institutions providing online students with disabilities accommodations that are comparable to those provided to students in traditional in-person degree programs? To explore this question, we compared students in a fully online biology degree program to students in the equivalent in-person degree program at a large research university. For each group, we assessed the frequency with which students register with the disability resource center, the range of specific accommodations provided, and course grades. Results show that students in the in-person program were nearly 30% more likely to be enrolled with the disability resource center, and that students in the online program were offered a narrower range of accommodations. However, in relative terms (i.e., compared to students without disabilities in their degree program), online students with disabilities perform better than in-person students with disabilities.

## 1. Introduction

Legal requirements institutionalized the provision of learning accommodations for students with disabilities in American colleges and universities [1–3]. Within this context, a disability is defined as "a physical or mental impairment that substantially limits one or more major life activities, a record of such impairment, or being regarded as having such an impairment" [1, 3]. Accommodations are an adjustment to a course or degree requirements made to allow a student with a disability to have equal access to that course or degree and, by definition, are intended to ensure that students with disabilities have educational experiences as equivalent as

possible. For these reasons, it is not possible to publicly release the raw data. Qualified researchers may request access through https://uoia.asu.edu/contact.

**Funding:** This work was supported by grant #GT11046 from the Howard Hughes Medical Institute (www.hhmi.org), awarded to JPC, SEB, PL, and ADA and grant #2012998 and #1644236 from the National Science Foundation (www.nsf.gov), awarded to SEB. The funders had no role in study design, data collection and analysis, decision to publish, or preparation of the manuscript.

**Competing interests:** The authors have declared that no competing interests exist.

possible to students without disabilities. However, as Gin et al. [4] discuss, contemporary higher education has changed dramatically since these statutes were enacted. Notably, because university structures that provide disability accommodations predate the availability and growth of both online courses and fully online degree programs, common accommodations provided are specific to the obstacles to learning that students might face when attending in-person courses. Therefore, in-person accommodations may not be as well suited to addressing the needs of online students with disabilities.

Online education offers some inherent accommodations relative to in-person settings, particularly when viewed through the historical lens of disability accommodation [e.g., 5]. For example, students do not need to physically move across campus to get to class, rendering some accommodations related to mobility moot. Similarly, pre-recorded lectures or video conference-based instruction easily allow for pausing, repeated viewing, or playback at different speeds. These can be seen as "built in" accommodations for students with certain conditions such as ADHD. Asynchronous instruction gives students greater time flexibility for completing their work, which can be helpful for students with chronic health conditions who may have frequent doctor appointments or flare ups.

At the same time, the modality of teaching online may present novel challenges and, thus, the need to consider how to adapt common accommodations or create new accommodations to best support fully online students. For example, test-taking accommodations, such as reduced distraction environments in a room on campus, cannot be provided for students who are physically situated across the world. Depending on students' living situations, a quiet testing environment may not be available. The issue of video-monitored exam proctoring has been increasingly debated during the COVID-19 pandemic [e.g., 6–8], with some work suggesting that it may exacerbate student mental health struggles [9–11].

Thus, it is an open question whether the inherent accommodations of the online modality allow students with disabilities to learn and succeed academically or if the range of accommodations offered to online students is, indeed, narrower and if this in turn hinders the performance of online students with disabilities. Please note that we chose to use "person-first language" (e.g., students with disabilities) in this article, although we do recognize that this choice is not universally preferred [9].

## 1.1. Access to disability accommodations for online students

A pair of recent studies gives some insight into the challenges faced by online students with disabilities. Gin et al. [9] interviewed students with disabilities in courses that were rapidly shifted to an online format in response to the COVID-19 pandemic. Gin et al. [4] conducted a follow-up survey a year later to test if students with disabilities were being provided adequate accommodations online after instructors had more time to be comfortable teaching online. In both studies the authors found that students with disabilities faced obstacles to receiving the disability accommodations to which they were legally entitled. Early on in the pandemic, students with disabilities were often completely forgotten about and their standard accommodations were often not enacted. A year later, students with disabilities were receiving their accommodations, but often these accommodations were not meeting their needs, both in the new modes of instruction and because of the changing needs of students due to the pandemic. Collectively, these studies highlight that students with disabilities currently are not being adequately supported in many online environments.

These studies brought important issues to light and, through the interviews and open-ended survey questions, allowed students with disabilities to reveal barriers associated with online learning in their own words. However, both studies are limited in that they only capture

the experiences of those students who chose to participate in the studies. By examining administrative data, the present study will build on Gin et al. [4, 9] and explore the experiences of all registered students with the Disability Resource Center at a single institution.

## 1.2. Academic performance of students with disabilities

To our knowledge, there are no prior studies that examine the academic performance of postsecondary students with disabilities in fully-online degree programs. In research exploring traditional in-person postsecondary settings, Kimball et al. [12] studied both persistence and academic achievement of students with disabilities. Although many results point to lower persistence, Kimball et al. argue that the evidence is not conclusive, owing generally to the use of correlational data. Similarly, Fichten et al.'s [13] review of existing evidence finds mixed conclusions, with studies finding equivalent persistence [14], albeit with longer time to graduation [15, 16], or less persistence [17, 18]. Looking at academic achievement, studies often examine the success of students with specific types of disabilities. For example, Dong & Lucas [19] examined academic performance of students across majors with a range of disabilities and, importantly, studied the performance of students who did and did not register for campus disability services. These authors found the students with disabilities—whether psychological, cognitive, or physical disabilities—were less likely to persist than students who reported no disability. They also found that students with psychological or cognitive disabilities who requested accommodations were more likely to be in good academic standing, although this relationship was not found for students with physical disabilities. Interestingly, Lee [20] found that STEM majors with disabilities received significantly fewer accommodations than non-STEM majors.

Although research has shown many specific disability accommodations to have a positive impact on student success, these results are not universal. A recent randomized controlled study examined the value of accommodations for students with ADHD or learning disabilities in which students were allowed to complete tests in a separate, reduced-distraction environment [21]. Their results show that not only did the separate testing room not improve test performance, but that students with ADHD or learning disabilities performed worse in the separate testing room compared to students in the classroom. Using the Beginning Postsecondary Students Longitudinal Study (BPS:04/06) data set, Mamiseishvili & Koch [22] studied factors that predicted first- to second-year persistence among students with disabilities, including specific accommodations. Analyzed in isolation, they found that classroom note-taking accommodation was significantly related to increased persistence and that alternative exam formats and additional time were not significant. However, these did not rise to the level of inclusion in the authors' final regression model. In a single-university study modeling cumulative GPA, Kim & Lee [23] found that including specific disability accommodations added only a small amount of explanatory power to their regression model. Newman et al. [24] looked more broadly and examined the effect on the persistence of students with disabilities of the use of resources that are available to all students regardless of disability status, such as tutoring and writing or study centers. Their results show that accessing only these universally available resources led to significantly higher persistence, whereas accessing only disability-related support had no effect on persistence. Notably, Newman et al. relied on data from the National Longitudinal Transition Study-2, a nationally representative study, and, thus, included students with disabilities who choose not to disclose this information to their college or university [25].

## 1.3. The present study

We examine administrative data from students in both an in-person and a fully online biology degree program at a large, public research university. Our focus on a science degree program

follows from the substantial body of work, particularly in recent years, showing failures in achieving diversity, equity, and inclusion in the sciences and engineering [e.g., 26–28] and our own prior work examining course grade equity for women, racial and ethnic minorities, low-income, and first generation to college students in online biology [29, 30]. Thus, against this backdrop, we consider whether students with disabilities in an online biology degree program are afforded an equitable experience both relative to their online peers without disabilities and relative to in-person degree students. In this study we pose the following research questions:

RQ1: Do in-person and fully online students differ in either the frequency of reported disabilities or the frequencies of receiving specific accommodations?

RQ2: Do students with disabilities compared to students without disabilities differ in academic performance between in-person and fully online degree programs?

## 2. Methods

### 2.1. Description of population and data sources

We collected three types of student data:

1. Academic data: individual course grades, overall grade point average,

2. Demographic data: student gender, race/ethnicity, college generation status, and Pell Grant eligibility (an indicator of socioeconomic status), and

3. Disability data: categorical disability type and specific accommodations requested by course.

All students were enrolled in the Biological Sciences degree program. This degree is offered both in-person and in a fully online mode, but both modes are housed in the same academic unit and were designed to be identical in their curriculum structure. We included course enrollments from Fall 2014–Fall 2019 for the in-person program and Fall 2017–2019 for the online program (Note that the online degree program began in Fall 2017, but grew rapidly in enrollment, eventually surpassing in-person enrollment). The end point was chosen to avoid the confounding effects of the COVID-19 pandemic, which necessitated a shift to remote instruction for all students beginning midway through the Spring 2020 semester. We do wish to acknowledge the effects the pandemic has had on students with disabilities; please see Gin et al. [4, 9] for examinations of those effects. In order to make our findings more general and to avoid undue influence from unique circumstances that can emerge in smaller courses, we limited our analysis to the large, required courses that are the focus of the first two years of the degree program. These include the two-course introductory biology sequence, genetics, evolution, a two-semester introductory chemistry series, two organic chemistry courses, and introductory physics. Most of these courses include a laboratory component. These are also the same set of courses analyzed in a previous study, the focus of which was course grade equity in online courses with respect to gender, race/ethnicity, household income, and college generation status [29].

Data regarding student disability status and the specific accommodation requests are stored separately from ordinary academic and demographic data. For this reason and to ensure there was no possibility that personally identifying information related to disability status was revealed, we took steps to ensure that the identifiable disability data were handled only by staff members within the Disability Resource Center (DRC). Note that we will use the DRC abbreviation as a generic term, but such organizations may also be called a Disability Services Office, Student Accessibility Center, among other names. The lead author compiled the academic and demographic data based on the selection criteria described above. These data were then sent to

the office in charge of approving and coordinating disability accommodations who performed a match to their internal database, de-identified the data, and returned the new dataset to the lead author for analysis. Details of this process were reviewed and approved by the Arizona State University institutional review board (IRB, protocol #9105). Consent was not obtained because the data were analyzed anonymously.

Prior research on the subject of disability accommodations has argued for the importance of including and prioritizing the perspectives of students with disabilities themselves [31]. The present study relies on de-identified administrative data. It would not be possible to conduct such a broad survey of the types of accommodations sought and the course grades earned by students with disabilities in these degree programs. Nonetheless, it is important to acknowledge that the administrative data do not capture the full depth and range of the academic experiences of these students and that since we are only analyzing students who are registered with the DRC, we are only examining the experiences of students who have the resources and support to have achieved a diagnosis.

## 2.2. Description of analyses

We calculated descriptive statistics for the student demographic variables, students' disability status, disability type, and the frequencies of disability accommodations that were received. The categories used for disability type are the same used in Gin et al. [4, 9, 31]. Following the procedure of our prior studies [29, 30], we used a linear mixed effects regression to estimate the effect of student disability status on course grades, adjusting for the effects of prior academic performance (GPA in other courses, abbreviated as GPAO, [27]), whether the student earned fewer than 30 credit hours, gender, race/ethnicity, age, college generation status, and socioeconomic status (fixed effects) and including random effects for each student and class section. GPAO was a continuous variable on a 0–4.33 (A+) scale. Age was treated as a categorical variable (18–25 and over 25 years of age). These categories distinguish the more "traditional" aged (18–25 years old) students from older students and are also of roughly equal sizes among the online program students. The remaining fixed effects were analyzed as binary variables: fewer than 30 credit hours or not, binary gender (female, male), race/ethnicity (BLNP [Black, Latine, Native American, or Pacific Islander], White or Asian), college generation status (first-generation, continuing-generation), socioeconomic status (Pell eligible, non-Pell eligible). We also used logistic regression to estimate the effects on DRC registration of degree program modality (online or in-person) and possible interaction effects between modality and gender, race/ethnicity, college generation status, and socioeconomic status. For model selections, we employed both forward selection (starting with a minimal model and adding predictors stepwise) and backwards elimination (starting with a full model that consisted of all of the above predictors and removing predictors stepwise) [32].

Note that, in contrast to Mead et al. [29], we did not fully exclude students with missing demographic data or students who received "withdraw" grades in a course. Because the focus of the present study goes beyond just grades analysis, there was no need reduce our analytical power by excluding these data when analyzing DRC enrollment or the types of accommodations given. However, for regressions involving grades or demographics, we excluded any course enrollments where the student received a "withdraw" grade and we excluded students with missing demographics data.

## 2.3. Positionality statement

Our research team consists of both women and men as well as first generation college graduates and individuals who received Pell grants as students. Some of us are members of the

LGBTQ+ community and some of us identify as having depression. Most of us have served as instructors of courses who have worked directly with the DRC to provide students with disabilities with accommodations. One of us has received accommodations for a disability through the DRC as an undergraduate and graduate student.

## 3. Results

### 3.1. Population demographics

The total population included 5586 students, 2908 from the in-person degree program and 2678 from the online degree program (Table 1). Women were a majority in both groups, although substantially more so in the online program (74% vs. 59%). About a third of students in both programs were BLNP. Just under half of the in-person students were Pell eligible, while somewhat more than half were Pell eligible among the online program students. Similarly, the percentage of first-generation students was also higher online (43% vs. 33%). In summary, although the two populations are similar, the online program has slightly more representation of each of the four historically marginalized groups (consistent with our own prior work, [29]). Another important demographic consideration is student age, which also differs substantially between the in-person and online populations. The median age for in-person students in our dataset is 19 as compared to 25 in the online population.

Table 2 shows the percentage of students in each program who are registered with the DRC. With 8% of in-person students and 4.7% of online students registered for a disability accommodation, this is well below published estimates for the overall proportion of students with a disability (19.4%; [33]). However, previous research also finds that only about a third of students with disabilities disclose this information to their school [25], which would put the

**Table 1. Student demographics by modality.**

|  | In-Person Students | Online Students |
|---|---|---|
|  | N = 2,908[1] | N = 2,678[1] |
| Gender |  |  |
| Man | 1,197 (41.2%) | 706 (26.4%) |
| Woman | 1,710 (58.8%) | 1,971 (73.6%) |
| Decline to state | 1 | 1 |
| Race/Ethnicity |  |  |
| White or Asian | 2,004 (69.6%) | 1,697 (64.4%) |
| BLNP | 876 (30.4%) | 938 (35.6%) |
| Decline to state | 28 | 43 |
| Socioeconomic Status |  |  |
| Non-Pell Eligible | 1,609 (55.3%) | 1,100 (41.1%) |
| Pell Eligible | 1,299 (44.7%) | 1,578 (58.9%) |
| College Generation Status |  |  |
| Continuing Generation | 1,944 (66.9%) | 1,518 (56.7%) |
| First-Generation | 964 (33.1%) | 1,160 (43.3%) |
| Age in Years (continuous) | 19 (18–21) | 25 (22–29) |
| Age in Years (categorical) |  |  |
| Age ≤ 25 | 2,682 (92.5%) | 1,515 (56.6%) |
| Age > 25 | 219 (7.5%) | 1,163 (43.4%) |
| Decline to state | 7 | 0 |

[1]n (%); Median (IQR); Note: percentages shown exclude "decline to state"

**Table 2. Frequency of "primary" disabilities by modality.**

| All Students | | |
| --- | --- | --- |
| | In-Person Students | Online Students |
| | N = 2,908[1] | N = 2,678[1] |
| No Accommodation | 2,676 (92.0%) | 2,552 (95.3%) |
| Any Disability Accommodation | 232 (8.0%) | 126 (4.7%) |
| **Students with any disability accommodation** | | |
| | In-Person Students | Online Students |
| Disability Type (with N $\geq$ 20) | N = 232[1] | N = 126[1] |
| Learning disability | 59 (25.4%) | 41 (32.5%) |
| Mental health/psychological disability | 100 (43.1%) | 44 (34.9%) |
| All others | 73 (31.5%) | 41 (32.5%) |

[1]n (%)

two populations in our study near to the prior estimates. Full demographic details for the students with any disability accommodation may be found in S1 Table. Table 2 also shows the percentages of students whose listed "primary" disability falls within either learning disability (including ADD/ADHD; see [9] for a discussion of this categorization) or mental health/psychological disability. These are the most common disability types in our data set, which is consistent with prior analysis [34]. Students in both groups were registered with other disabilities types, including, Acquired Brain Injury, Chronic health condition, Hearing loss, Neurological, Physical disability, and Visual loss, but each of these categories had fewer than 20 individual students and our research protocol prohibits us from presenting results for subgroups smaller than this size. It is also important to note that the personal experiences of individuals, even with the same type of disability, are unique [35, 36]. Thus, we caution against making generalizations concerning all individuals who share a disability type or specific disability.

### Finding 1: DRC enrollment is significantly lower among online program students

There are two important dimensions to this research question: differential access to (or use of) disability support services and a differential range of services provided. To examine the first dimension of this, we used logistic regression to determine whether students in the online or in-person modalities were equally likely to be registered with the university DRC. To test for possible differences within these populations, we performed additional regressions that included student demographics and interactions between the degree program type and each of gender, race/ethnicity, Pell grant eligibility, college generation status, age, and whether the student has fewer than 30 credit hours.

Overall, we find significant differences in DRC enrollment associated with degree program mode (in-person or fully online). Specifically, in-person program students are nearly 30% more likely to be enrolled with the DRC (Table 3). Regarding demographics, we will first consider a main effects model to examine how the demographic effects differ by degree program mode. We will then add a series of interaction terms to see whether these demographic effects vary in their impact for in-person and online students. In the main effects model (S2 Table), we find that women are much more likely to be registered with the DRC as are students older than traditional college age. Pell eligible students are slightly more likely to be registered while first-generation students are somewhat less likely to be registered. No significant differences exist in the main effects model with respect to race/ethnicity or credit hours earned.

**Table 3. Difference in DRC enrollment by degree program.**

| Characteristic | OR[1] | 95% CI[1] | p-value |
|---|---|---|---|
| (Intercept) | 0.06 | 0.06, 0.07 | <0.001 |
| Online | — | — | |
| In-Person | 1.29 | 1.14, 1.46 | <0.001 |

[1]OR = Odds Ratio, CI = Confidence Interval

Considering the full model with degree program modality interaction effects (Table 4), we see that women are more likely to be registered in both programs, but that this effect is stronger for in-person students. In contrast to the main effects model, we see that BLNP is significant when the program interaction effects are considered. BLNP students in the online program are more likely than white or Asian students to be registered with the DRC, but in the in-person program BLNP students are slightly less likely to be registered. There are also similar, but smaller differences with respect to college generation status, with first generation students in the in-person program being significantly less likely to be registered than first generation students in the online program. For online students, having fewer than 30 credit hours is a negative predictor of DRC registration; this is not the case in-person. No significant interactions with program modality were found for student age or Pell eligibility.

**Table 4. Difference in DRC enrollment by student demographics and degree program.**

| Characteristic | OR[1] | 95% CI[1] | p-value |
|---|---|---|---|
| (Intercept) | 0.03 | 0.02, 0.04 | <0.001 |
| Campus | | | |
| Online | — | — | |
| In-Person | 1.55 | 1.12, 2.18 | 0.009 |
| Gender | | | |
| Man | — | — | |
| Woman | 1.88 | 1.43, 2.49 | <0.001 |
| College Generation Status | | | |
| Continuing Generation | — | — | |
| First-Generation | 0.92 | 0.75, 1.14 | 0.5 |
| Socioeconomic Status | | | |
| Non-Pell Eligible | — | — | |
| Pell Eligible | 1.23 | 1.08, 1.40 | 0.002 |
| Race/Ethnicity | | | |
| White or Asian | — | — | |
| BLNP | 1.31 | 1.06, 1.62 | 0.013 |
| Age in Years | | | |
| Age $\leq$ 25 | — | — | |
| Age > 25 | 1.42 | 1.19, 1.68 | <0.001 |
| Fewer than 30 Credit Hours | 0.65 | 0.45, 0.92 | 0.019 |
| Campus Interaction Effects | | | |
| In-Person * Woman | 1.38 | 0.99, 1.89 | 0.053 |
| In-Person * First-Gen. | 0.77 | 0.59, 1.02 | 0.064 |
| In-Person * BLNP | 0.68 | 0.52, 0.90 | 0.006 |
| In-Person * Fewer than 30 Credit Hours | 1.56 | 1.04, 2.40 | 0.036 |

[1]OR = Odds Ratio, CI = Confidence Interval

## Finding 2: Online degree program students with disabilities are given access to a narrower range of accommodations

Table 5 presents an overall summary, for common accommodation categories, of the frequency at which they are received by students in both the in-person and online program courses. A complete list of accommodation types is provided in S3 Table. There are several accommodation "types" that are much less common for online students than in-person students. These include:

- "Reduced Distraction" environment for testing

- Flexible attendance

- Peer notetaking services

- Audio recording

The impact of these varies in severity. Depending on the nature of the online course, "audio recording" and "flexible attendance" may be irrelevant in the majority of cases. However, both "reduced distraction environment" and "peer notetaking services" are accommodations that can reasonably be seen as addressing needs that are common to both in-person and online learning. A previous nationwide study found notetaking to be the third most common accommodation with 26% of students surveyed reporting receiving this kind of support [18]. It is important to state that the notetaking accommodation here refers to a peer notetaker, i.e., a fellow classmate who is compensated to share their own notes with the student receiving the accommodation. Thus, while dictation software or other technology solutions have some overlap with the intended benefits of the peer notetaking services accommodation, those tools are not entirely equivalent.

Conversely, we see that some accommodations are somewhat more common in the online group. These include Assistive Technology and PDF with Recognized Text (i.e., ensuring that PDF documents are compatible with screen readers). These accommodations are understandably important in the computer-based learning environment of the online program. We found that extra time on exams is the most common accommodation in both modalities and flexible assignment deadlines is the second and fourth most common for online and in-person, respectively, but in both cases, the percentages of students receiving these accommodations are higher in the online modality.

**Table 5. Percentage of accommodation-eligible students receiving common[1] accommodations by mode.**

| Accommodation | In-Person, N = 270[2] | Online, N = 111[2] |
|---|---|---|
| Assistive Technology | 5 (1.9%) | 7 (6.3%) |
| Audio Recording | 21 (7.8%) | 1 (0.9%) |
| Extra Time on Exams | 71 (26.3%) | 44 (39.6%) |
| Flexible Assignment Deadlines | 27 (10.0%) | 32 (28.8%) |
| Flexible Attendance | 18 (6.7%) | 2 (1.8%) |
| Peer Notetaking Services | 38 (14.1%) | 0 (0.0%) |
| PDF with Recognized Text | 9 (3.3%) | 10 (9.0%) |
| Reduced Distraction | 49 (18.1%) | 4 (3.6%) |

[1] i.e., accommodations with at least 5% occurrence in one mode

[2] n (%)

### Finding 3: The relative performance of students with disabilities in the online program exceeds that of the in-person program

Our regression model finds a significant interaction effect between disability status (i.e., a student requesting a disability accommodation for a particular course) and learning modality. Specifically, in-person students with disabilities earn grades 0.19 grade units lower than their peers as compared to students in the online degree program (Fig 1, Table 6). The overall grade effect associated with disability accommodation was positive, but non-significant. This model also finds significant effects associated with demographics categories and online program status. Similar demographic findings from a closely related student population were described in [29]. Recognizing Finding 1—significant demographic and program modality differences in

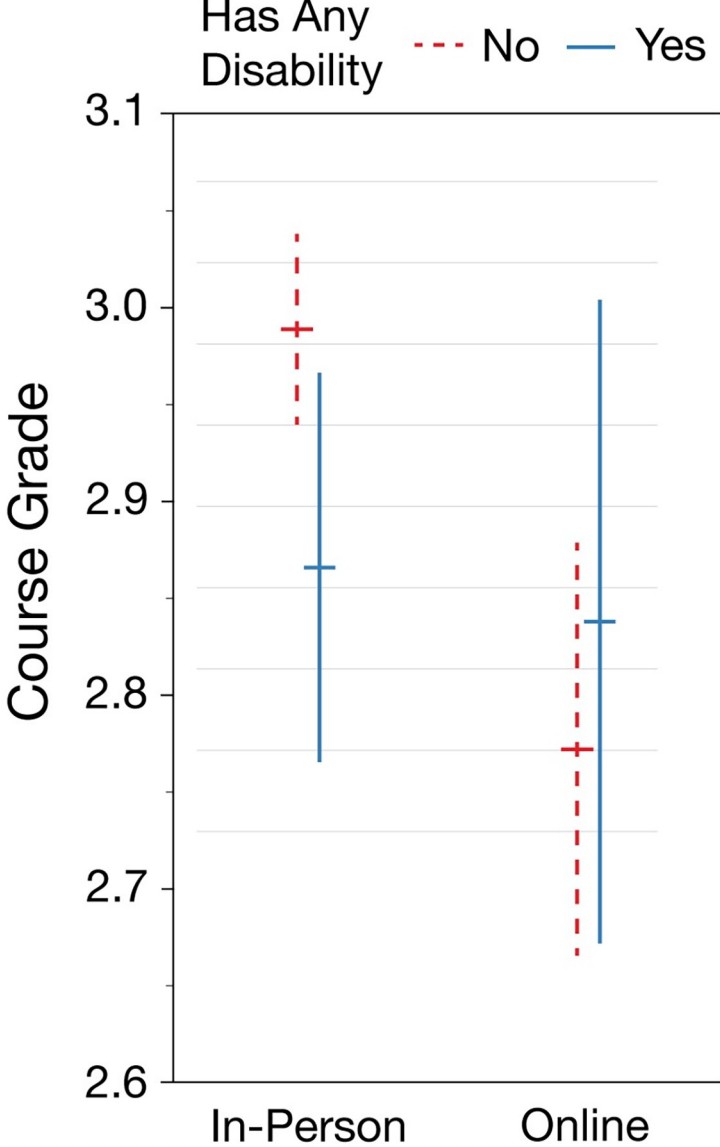

**Fig 1. Modeled grade effects for students by disability status and learning modality.** Whereas having a disability was not associated with an overall grade difference, the results do show a significant interaction effect between disability status and modality.

**Table 6. Regression results.**

| Characteristic | Beta | 95% CI[1] | p-value |
|---|---|---|---|
| (Intercept) | 0.63 | 0.50, 0.76 | <0.001 |
| GPAO | 0.69 | 0.66, 0.71 | <0.001 |
| Campus | | | |
| Online | — | — | |
| In-Person | 0.22 | 0.11, 0.33 | <0.001 |
| Gender | | | |
| Man | — | — | |
| Woman | -0.07 | -0.11, -0.03 | <0.001 |
| College Generation Status | | | |
| Continuing Generation | — | — | |
| First-Generation | -0.11 | -0.15, -0.07 | <0.001 |
| Socioeconomic Status | | | |
| Non-Pell Eligible | — | — | |
| Pell Eligible | -0.05 | -0.09, -0.01 | 0.008 |
| Race/Ethnicity | | | |
| White or Asian | — | — | |
| BLNP | -0.21 | -0.25, -0.16 | <0.001 |
| Age in Years | | | |
| Age ≤ 25 | — | — | |
| Age > 25 | 0.04 | -0.01, 0.09 | 0.10 |
| Fewer than 30 Credit Hours | -0.27 | -0.32, -0.22 | <0.001 |
| Has Any Disability | 0.07 | -0.07, 0.20 | 0.3 |
| Campus Interaction Effect | | | |
| In-Person * Has Disability | -0.19 | -0.35, -0.02 | 0.024 |

[1]CI = Confidence Interval

DRC enrollment—we explored the addition of interactions between these factors and the disability status term, but none of these interactions were statistically significant.

We explored possible interaction effects involving specific types of disabilities using the same categories as in Table 6. The *learning disability* and *mental health/psychological disability* categories represent a large majority of students in this population (see Table 2). Our modeling showed students in each of these categories to have similar patterns of course performance to our initial regression model (i.e., to have a negative grade effect associated with the in-person degree program). See S4 Table for details. The small number of students with other types of disabilities limited our ability to detect possible interaction effects associated with any of those disability types.

## 4. Discussion

Regarding our first research question, our results show that systematic differences exist between the two modalities of degree programs studied, with students in the online program being less likely to enroll with the DRC or request disability accommodations. There are also notable differences in the demographic effects by modality, such as online BLNP students being more likely to have a disability accommodation and online students with fewer than 30 credit hours being less likely to have any accommodation. The online students are also offered a narrower range of disability accommodations. With respect to our second question, we find

that the relative academic performance of students with disabilities to students without disabilities differs between the online and in-person degree programs. In relative terms (i.e., compared to students without disabilities in their degree program), online students with disabilities perform better than in-person students with disabilities.

The differences in the types of accommodations provided online as compared to in-person reflect a combination of accommodations that are impractical/impossible to provide to a distributed and remote population of students and accommodations that are inherently unnecessary online. This is very much the pattern we anticipated, and it highlights the inherent advantages and disadvantages of online learning for students with disabilities. However, the differential rates of registration and requests for accommodations with the university DRC across both modality and student demographics raise questions about whether all students are being made aware of and given access to these resources. Our findings are consistent with the issues raised in Gin et al. [4] and Terras et al. [37], both in the specific lack of access to distraction-free testing and peer notetaking services for online students and in the overall lower rates of DRC registration among the online students studied.

Our findings with respect to the demographic predictors of DRC enrollment contribute to a somewhat varied set of previous findings. The largest demographic effect we observed was that of gender, in which women were much more likely to be registered for an accommodation. This sits in contrast to Henderson [38], Wagner et al. [39], and Newman et al. [25] which present evidence of the opposite trend. However, the U.S. Department of Education reported gender parity with respect to disability status among undergraduates and found that women were more likely to report a disability among postbaccalaureate students [33]. It is important to note that our study population has a high population of women, owing in part to the discipline of the program studied (biology) and in part to the fact that the online program enrolls proportionately more women [29]. As additional studies are performed involving online programs, it will be interesting to see how these demographic effects compare to results from in-person programs.

It goes beyond our data to make claims about whether the underlying rate of disabilities differs systematically between the in-person and online degree programs. However, if we assume that this rate of disability is constant, then our data point to systematic differences between these two degree programs across one or more of a number of factors related to how students with disabilities approach these programs. This may include students' **awareness** of these university services or of their personal **eligibility** for receiving them. It may include students' perceived **value** of the available accommodations or their **willingness to request** accommodations. Lastly, the differences in usage may stem from perceived and real differences in the **need** for accommodation in the in-person versus online programs, even for students with very similar personal circumstances. We expand upon each of these possibilities in the following paragraphs.

## 4.1. Awareness

It is possible that the online program students are less well-informed about the availability of support through the DRC [e.g., 9, 40]. This could follow from a lack of informal sharing of information that is more likely to occur in in-person learning environments. Supporting this explanation is the fact that for online students, having fewer than 30 credit hours was predictive of *less* DRC enrollment, whereas this was not the case among in-person students. This suggests that, despite the university's many lines of communication to its online program students, including traditional academic advisors and "success coaches" who provide support to online students for things like time management and career exploration, many students early in their college journey may not receive the support that they may require and be entitled to.

## 4.2. Eligibility

Complicating this subject is the question of which students are considered eligible to receive accommodations. In addition to the structural issues addressed in the Gin et al. studies, prior research has highlighted the "documentation disconnect", in which a student was deemed eligible for a disability accommodation at the K–12 level, but, due to more stringent requirements for documentation of disability, was not found to be eligible for the same accommodation at the college level [41, 42]. Sparks & Lovett [43] also conclude that the breadth of methods for diagnosing a learning-disabled student has led to a situation in which there is substantial overlap in the academic performance of "learning disabled" and "non-learning disabled" students. The literature calls attention to ways that a student may have an expectation of receiving a disability accommodation, but not be eligible in practice. Some of these factors may be exacerbated in the case of fully online degree programs. For example, the documentation disconnect described previously occurs in part because different laws mandate disability accommodation in K–12 than in higher education and in part because standards for K–12 disability status vary by state. Given that online undergraduate programs are often marketed toward out-of-state students, the fraction of these ineligible students may be greater in an online program as compared to the traditional in-person degree programs at the same university. In addition, given that our prior work showed that the online program attracted relatively more students from lower socioeconomic status backgrounds [29], it is possible that online students with disabilities are, on average, less able than the in-person degree students to obtain the medical diagnoses necessary to demonstrate their eligibility.

## 4.3. Value

Assuming that online students with disabilities are aware of their support options and, bearing in mind our results showing the limited range of accommodations that are commonly received (Table 5), it is possible that some students are making an informed choice to not ask for accommodations. That is to say that these students may believe that the accommodations that are made available to them do not effectively address their needs. We have no evidence that speaks directly to this possibility. However, in considering more indirect evidence, Gin et al. [4, 9] found that some students with disabilities struggled to be granted the kind of support they felt was justified during the emergency shift to online learning during the COVID-19 pandemic. There is also the question of a perceived stigma associated with requesting accommodations, so students must see these accommodations as having a value that exceeds the effort required to obtain them and any negative consequences (e.g., judgement or bias against them) they may associate with them.

## 4.4. Willingness to request

Numerous previous studies highlight the important possibility of students who choose not to address or report their disabilities [5, 12, 25, 44]. This may be even more true in online courses and programs where students will have fewer and more limited interactions with their peers or with faculty members. The limited nature of these interactions makes it easier, and perhaps more appealing, to keep one's disability status private. It also removes opportunities for students to learn about the types of accommodations available or the potential value of those resources. Similarly, it may be more difficult for online instructors (or student peers) to recognize instances where a student may benefit from disability services. Therefore, the mediated nature of an online degree program expands the concept of a "hidden disability", providing many students with disabilities the choice whether or not to disclose their disability status to others. In an in-person setting, a hidden disability might be a mental health condition, but in

an asynchronous online setting, this category could include deafness, physical disabilities, or chronic health conditions, many of which would be readily apparent in-person. Thus, students not registering with the DRC could be a conscious decision not to reveal their disability to others, and the nature of online learning gives students in those degree programs more autonomy regarding this decision.

## 4.5. Need

Looking at results from both Tables 5 and 6, a final possibility is that the differences in the frequencies of requested accommodations follow from the inherent accommodations provided by the online learning modality. Put simply, perhaps students online need fewer accommodations because of the asynchronous flexible nature of the learning environment. Consider peer notetaking services, for example, which our results show to be a relatively common accommodation in-person, but not available to online students. It may be the case that some students who would have requested this kind of accommodation for a synchronous, in-person class do not see it as necessary for an asynchronous, online delivery where they can freely pause, rewind, or rewatch lectures at their convenience. Alternatively, perhaps our results are driven by a self-selection effect in which students with disabilities preferentially enroll in the online degree program knowing that they will not need to request disability accommodations. This kind of strategic enrollment would imply a high level of effective self-determination, something that prior work has shown to be associated with academic success [45]. Therefore, if such behavior is widespread, then our observed grade differences reflect a combination of the inherent affordances of the online modality and the presence of students with the skills of self-determination that help them to be successful.

Prior research is mixed on whether online courses are seen as preferable by students with disabilities, with much of the difference coming from how attentive a given institution or instructor has been to accessibility [12, 46, 47]. The present study does not examine instructional practices or technology use at the level of individual courses, but strategies for effective and accessible online learning have been reviewed elsewhere [e.g., 48] and could be the basis for future research expanding on our work.

This study relied on administrative data because these data provide a complete summary of the university's available DRC services and students' use of these services. However, our work cannot speak directly to the students' perspective in requesting, declining to request, or receiving DRC services, nor can it speak to factors related to the self-determination of these students. The latter is one of the more widely studied constructs for both predicting success of students with disabilities and for designing support programs to promote success [e.g., 49–52]. Future work is needed to explore whether our findings reflect an underlying difference in the level of self-determination between students in in-person and online degree programs or, perhaps, that the skills associated with self-determination (self-advocacy, goal setting, etc.) must be applied differently in online settings.

We conclude this section by reiterating that our regression results with respect to course grades suggest that students with disabilities who are registered with the DRC in the online degree program have an equal or better opportunity to succeed as their in-person counterparts. Therefore, we tentatively conclude that online students with disabilities can be well-supported in that modality. However, we do underscore that our interpretation of the grade results is complicated by the fact that we can only analyze students who were officially eligible for *and* proactively chose to request support from the campus DRC, thus it may be the case that our finding with respect to grades is biased by a selection for the most well-informed students with disabilities. Or, relatedly, that online students with disabilities that are registered

with the DRC are the more privileged group of students with disabilities, so the grade advantage that we see online is simply because the more privileged group of students with disabilities are represented in the dataset.

## 4.6. Limitations

Although we believe this work is an important first step in closing a gap in the existing understanding of disability accommodations in online learning environments, we also wish to highlight some limitations. First, there are reasons to predict that some students with disabilities might be more likely to prefer an online degree program. This could lead the population of students with disabilities online to be systematically stronger academically and more motivated to succeed. Testing this possibility would require an indicator of prior academic performance, such as high school GPA or standardized test scores, but these data are not uniformly collected at admission to the online degree program that we studied, thus we are unable to rule it out. Second, the overall percentage of students with disabilities is smaller in the online program than the in-person one. If there exist substantial numbers of online students who could benefit from disability accommodation, but who are not registered to receive them, this could have biased our comparison of online to in-person grades by disability accommodation status.

Although we explored the possible differential effects among students with different types of disabilities and found no such differences, it bears repeating that our primary results aggregate all students with any disability. It goes without saying that the nature of the barriers to academic achievement experienced by a student with reduced mobility and those experienced by a student with a learning disability are very different. The same is true within these broad categories of disability. Critically, we also cannot assume that an in-person student and an online student with the same type of disability will have the same barriers to academic success. It is also important to note that the personal experiences of individuals, even with the same type of disability, are unique [35, 36]. Thus, we caution against making generalizations concerning all individuals who share a disability type or specific disability and acknowledge that our aggregated analyses may conceal important variability.

The other notable limitation is the use of administrative data. Although these data did allow us to examine research questions that are troublesome for survey and interview research-based approaches, the administrative data do not capture a complete picture of any one student's experience. This is particularly true when studying students with disabilities, each of whom must be categorized within an existing category for disability type (and other demographic characteristics).

## 5. Conclusion

The use of online learning will certainly continue to grow among institutions of higher education. It is, therefore, essential that these institutions examine and continuously monitor how their existing disability accommodations align with the needs of students in online courses and fully online degree programs. Previous survey- and interview-based research has found that students with disabilities in online courses feel less well supported and encounter more obstacles to receiving accommodations [4, 9, 37]. In our study, administrative data from a fully online degree program suggests that this kind of unequal accommodation persists. While our analysis of course grades indicates that the affordances of online learning for students with disabilities may outweigh any disadvantages caused by the gaps in accommodation, there remains an obligation for administrators and faculty to ensure that students are equitably supported across both in-person and online programs. In particular, if the types of accommodations offered predate the online program, there may be gaps either due to the appropriateness of

those accommodations for fully online courses or due to the practical realities of providing those accommodations to remote students. Although the details of disability accommodations will vary, we hope that the present study will offer a starting point for self-study at any institution with new or existing online degree programs and that our results will inspire these institutions to look for ways to better support their online students with disabilities.

## Supporting information

**S1 Table. Student with disability's demographics by modality.**
(DOCX)

**S2 Table. Difference in DRC enrollment by student demographics.**
(DOCX)

**S3 Table. Complete list of accommodations in dataset.**
(DOCX)

**S4 Table. Regression results for disability type.**
(DOCX)

## Acknowledgments

We thank ASU's Student Accessibility and Inclusive Learning Services for their support in providing access to the anonymous data analyzed in this study.

## Author Contributions

**Conceptualization:** Chris Mead, Ariel D. Anbar, James P. Collins, Paul LePore, Sara E. Brownell.

**Data curation:** Chris Mead, Chad Price.

**Formal analysis:** Chris Mead.

**Funding acquisition:** Ariel D. Anbar, James P. Collins, Paul LePore, Sara E. Brownell.

**Investigation:** Chris Mead, Sara E. Brownell.

**Methodology:** Chris Mead.

**Project administration:** Chris Mead, James P. Collins, Sara E. Brownell.

**Supervision:** Sara E. Brownell.

**Validation:** Chris Mead, Chad Price, Logan E. Gin, Sara E. Brownell.

**Visualization:** Chris Mead.

**Writing – original draft:** Chris Mead.

**Writing – review & editing:** Chris Mead, Chad Price, Logan E. Gin, Ariel D. Anbar, James P. Collins, Paul LePore, Sara E. Brownell.

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
