## [Decision Letter · Decision Letter 0]

2 May 2023

PONE-D-23-10145

A comparative case study of the accommodation of students with disabilities in online and in-person degree programs

PLOS ONE

Dear Dr. Mead,

Thank you for submitting your manuscript to PLOS ONE. After careful consideration, we feel that it has merit but does not fully meet PLOS ONE’s publication criteria as it currently stands. Therefore, we invite you to submit a revised version of the manuscript that addresses the points raised during the review process.

We look forward to receiving your revised manuscript.

Kind regards,

Jolanta Maj

Academic Editor

PLOS ONE

Journal Requirements:

Additional Editor Comments:

The reviewers agreed on the high quality of the submitted manuscript and cited its importance. The reviewers pointed out possible corrections that would improve the text, which I, as editor, also agree with. Therefore, I would like to ask you to address the reviewers' comments or argue why making changes is not possible. 

Reviewers' comments:

Reviewer's Responses to Questions

**Comments to the Author**

1. Is the manuscript technically sound, and do the data support the conclusions?

Reviewer #1: Yes

Reviewer #2: Yes

2. Has the statistical analysis been performed appropriately and rigorously? 

Reviewer #1: Yes

Reviewer #2: Yes

3. Have the authors made all data underlying the findings in their manuscript fully available?

Reviewer #1: No

Reviewer #2: No

4. Is the manuscript presented in an intelligible fashion and written in standard English?

Reviewer #1: Yes

Reviewer #2: Yes

5. Review Comments to the Author

Reviewer #1: Dear authors,

I have had the privilege of reviewing your paper titled "Comparing Disability Accommodations in Online and In-Person Biology Degree Programs". As a reviewer, my goal is to help improve the quality of the paper so that it is suitable for publication in the PLOS ONE Journal. To this end, I have provided a detailed evaluation of your work in the attached "Reviewer's Comments" file. I hope these observations will be useful to you in reviewing and improving your manuscript. Please note that my comments are offered with the intention of being constructive and improving the quality of the work.

Best regards,

Reviewer #2: The authors made the analysis of administrative data to compare the frequency of reported disabilities, frequencies of receiving specific accommodations and academic performance of in-person and fully online degree programs students. I really appreciate the work which was done to minimize the effect of external factors, including COVID-19 effects. According to my opinion, both the research and the paper are very well prepared, therefore my remarks are rather not very significant:

1. Please explain the inconsistence in the information in Ethics Statement (Consent was not obtained because the data were analyzed anonymously. The work was done under a protocol approved by the ASU IRB. This is stated in the manuscript) and Data Availability Section (The data analyzed in this study involve students' individual disability statuses and are considered very sensitive. We were granted permission to analyze them, but do not have permission to disseminate the data themselves). This is explained in details in the paper (line 182 and next), but need to be clarify in the Additional information section.

2. There is a huge discrepancy in number of students using notetaking services (Table 5. Percentage of accommodation-eligible students receiving common accommodations by mode - line 344). Did you check if the reason is using notetaking (dictating) software instead notetaking services by online students?

3. I really appreciate the analysis of factors related to how students with disabilities approach programs (line 412 and next), however in the part Willingness to request (line 471 and next) please consider if one of the reason is lover pressure of the environment during online classes. Visibility of students difficulties and needs during online classes is much more difficult to recognize by both teachers and peers and therefore fostering to disclosure is significantly lower in this group.

4. Another possible factor to analyze is higher motivation of some students with disabilities to study online because it seems to be easier option to them. Are students with ASD more willing to online studies because of their communication difficulties? Are students with motor difficulties or blindness more willing to online studies because of their mobility difficulties?

5. Please check „orphans” in text editing, i.e. line 318-319, 381, 389.

6. PLOS authors have the option to publish the peer review history of their article (what does this mean?). If published, this will include your full peer review and any attached files.

Reviewer #1: No

Reviewer #2: **Yes: **Katarzyna Jach

---

## [Author Response · Author response to Decision Letter 0]

18 May 2023

Please see response to reviewers in the uploaded document.

---

## [Decision Letter · Decision Letter 1]

4 Jul 2023

A comparative case study of the accommodation of students with disabilities in online and in-person degree programs

PONE-D-23-10145R1

Dear Dr. Mead,

We’re pleased to inform you that your manuscript has been judged scientifically suitable for publication and will be formally accepted for publication once it meets all outstanding technical requirements.

Kind regards,

Jolanta Maj

Academic Editor

PLOS ONE

Additional Editor Comments (optional):

Thank you for the opportunity to read you excellent paper! All the necessary changes have been made! Congratiulations.

Reviewers' comments:

Reviewer's Responses to Questions

**Comments to the Author**

1. If the authors have adequately addressed your comments raised in a previous round of review and you feel that this manuscript is now acceptable for publication, you may indicate that here to bypass the “Comments to the Author” section, enter your conflict of interest statement in the “Confidential to Editor” section, and submit your "Accept" recommendation.

Reviewer #1: All comments have been addressed

Reviewer #2: All comments have been addressed

2. Is the manuscript technically sound, and do the data support the conclusions?

Reviewer #1: Yes

Reviewer #2: (No Response)

3. Has the statistical analysis been performed appropriately and rigorously? 

Reviewer #1: Yes

Reviewer #2: (No Response)

4. Have the authors made all data underlying the findings in their manuscript fully available?

Reviewer #1: Yes

Reviewer #2: (No Response)

5. Is the manuscript presented in an intelligible fashion and written in standard English?

Reviewer #1: Yes

Reviewer #2: (No Response)

6. Review Comments to the Author

Reviewer #1: (No Response)

Reviewer #2: (No Response)

7. PLOS authors have the option to publish the peer review history of their article (what does this mean?). If published, this will include your full peer review and any attached files.

Reviewer #1: **Yes: **Víctor-Hugo Perera Rodríguez

Reviewer #2: No

---

## [Editor Report · Acceptance letter]

7 Jul 2023

PONE-D-23-10145R1 

A comparative case study of the accommodation of students with disabilities in online and in-person degree programs 

Dear Dr. Mead:

I'm pleased to inform you that your manuscript has been deemed suitable for publication in PLOS ONE. Congratulations! Your manuscript is now with our production department. 

Kind regards, 

on behalf of

Dr. Jolanta Maj 

Academic Editor

PLOS ONE